# Anti-Ovarian Cancer Conotoxins Identified from *Conus* Venom

**DOI:** 10.3390/molecules27196609

**Published:** 2022-10-05

**Authors:** Shuang Ju, Yu Zhang, Xijun Guo, Qinghui Yan, Siyi Liu, Bokai Ma, Mei Zhang, Jiaolin Bao, Sulan Luo, Ying Fu

**Affiliations:** 1Key Laboratory of Tropical Biological Resources of Ministry of Education, School of Pharmaceutical Sciences, Hainan University, Haikou 570228, China; 2Beijing Key Laboratory of Organic Materials Testing Technology & Quality Evaluation, Institute of Analysis and Testing, Beijing Academy of Science and Technology, Beijing 100094, China; 3Medical School, Guangxi University, Nanning 530004, China

**Keywords:** anti-ovarian cancer, disulfide connectivity, disulfide-rich conotoxins, voltage-gated sodium channel, *Conus* venom

## Abstract

Conotoxins constitute a treasury of drug resources and have attracted widespread attention. In order to explore biological candidates from the marine cone snail, we isolated and identified three novel conopeptides named as Vi14b, Vi002, Vi003, three conotoxin variants named as Mr3d.1, Mr3e.1, Tx3a.1, and three known conotoxins (Vi15a, Mr3.8 and TCP) from crude venoms of *Conus virgo*, *Conus marmoreus* and *Conus texile.* Mr3.8 (I-V, II-VI, III-IV) and Tx3a.1 (I-III, II-VI, IV-V) both showed a novel pattern of disulfide connectivity, different from that previously established for the µ- and ψ-conotoxins. Concerning the effect on voltage-gated sodium channels, Mr3e.1, Mr3.8, Tx3a.1, TCP inhibited Na_v_1.4 or Na_v_1.8 by 21.51~24.32% of currents at semi-activated state (TP2) at 10 μmol/L. Certain anti-ovarian cancer effects on ID-8 cells were exhibited by Tx3a.1, Mr3e.1 and Vi14b with IC_50_ values of 24.29 µM, 54.97 µM and 111.6 µM, respectively. This work highlights the role of conotoxin libraries in subsequent drug discovery for ovarian cancer treatment.

## 1. Introduction

When comparing with other types of cancer, ovarian cancer is the eleventh most common in women, the fifth-most common cause of cancer-related death in women, and the most fatal gynecologic malignant cancer for females [1,2,3]. Hidden deep and often known as a silent killer, ovarian cancer is frequently not diagnosed (over 70%) until the disease has advanced to the late stage, since its symptoms are vague and difficult to detect [4,5]. In the year 2022 alone, 19,880 new cases and 12,810 deaths related to ovarian cancer were estimated by the National Cancer Institute of the United States [3]. Treatment of ovarian cancer is generally performed by a combination of debulking surgery and subsequent chemotherapy with the increasing use of immunotherapy and targeted therapies such as bevacizumab and PARP (poly ADP-ribose polymerase) inhibitors [6,7,8]. Paclitaxel/carboplatin and doxorubicin/carboplatin regimens are often used as first-line chemotherapeutic agents [2]. However, failure of chemotherapy may occur due to drug resistance problems and toxicity [9]. Herein, alternative anti-ovarian cancer drug candidates are urgently needed for the treatment of ovarian cancer.

Conotoxins (Ctx) are a class of active polypeptides secreted by cone snails (a carnivorous mollusk living in tropical oceans) and are mainly used for prey capture and defense against natural enemies [10,11,12]. Conotoxins usually consist of 10–40 amino acids and contain 0–8 cysteine (Cys) residues, which can form 0–4 pairs of intramolecular disulfide bridges. Conotoxins can specifically target on diverse crucial receptors (ligand-gated ion channels, voltage-gated ion channels, G protein-coupled receptors, and neurotransmitter transporters, etc. [13,14,15]) related to tough diseases, such as neuralgia, addiction, epilepsy, mental illness, cancer and other intractable diseases [16,17,18,19]. To date, more than seven contoxins, including ω-MVIIA (chronic pain, marketed), ω-CVID (analgesia, phase IIa), contulakin-G (analgesia, phase I), conantokin-G (analgesia/anti-epileptic, phase Ib), χ-MrIA (analgesia, phase IIa), α-Vc1.1 (analgesia, phase II, terminated), μO-MrVIB (analgesia, phase II), etc., have advanced to the clinical research stage [20,21]. Conoxin ω-MVIIA, which targets N-type human Ca_v_2.2 channels, has been developed as an FDA-approval analgesic drug Ziconotide (Prialt) for severe chronic pain since its launch in 2004 [22]. Cone snail venom is a huge drug resource, since it is estimated that cone snails can generate up to one million different bioactive peptides [23]. However, less than 0.1% of these useful peptides have been structurally and functionally characterized [15,18,21]. Herein, conotoxin has attracted extensive attention and becomes a hot spot in new drug development.

In order to explore biological candidates from marine cone snails, we isolated and identified three novel conopeptides named as Vi14b, Vi002, Vi003, three conotoxin variants named as Mr3d.1, Mr3e.1, Tx3a.1, and three known conotoxins (Vi15a, Mr3.8 and TCP) from crude venoms of *Conus virgo*, *Conus marmoreus* and *Conus texile* collected in the South China Sea. In this work, we reported their isolation, sequence identification, synthesis, disulfide connectivity recognition and biological evaluation.

## 2. Results and Discussions

### 2.1. Peptide Isolation and Sequencing

Crude venom powders weighing 4.1 mg, 18.9 mg and 11.6 mg were prepared from the venom dusts of *C. virgo*, *C. marmoreus* and *C. texile* snail specimens, respectively. Through systemic separation, three novel conopeptides named as Vi14b, Vi002, Vi003, three conotoxin variants named as Mr3d.1, Mr3e.1, Tx3a.1, and three known conotoxins (Vi15a, Mr3.8 and TCP) were isolated and identified from the crude venoms of these three species of cone snails (Table 1).

The alkylation result of the isolated conopeptides showed that none of them contained free thiol. The mass difference (4.03 Da, Appendix A) between the intact and the reduced Vi14b indicated that it contained two pairs of disulfide bridges. The Edman degradation result (Appendix A) indicated a sequence of QQM*****GR*****GGTGQII*****NE*****KT*****HGK*****V****.** The matrix-assisted laser desorption-ionization–time of flight-tandem mass (MALDI-TOF-MS/MS) spectrum of Vi14b was shown in Figure 1. The serial *b*_1_–*b*_2_, *b*_6_–*b*_7_, *b*_9_, *b*_11_–*b_1_*_2_, *b_1_*_4_, *b*_15_–*b*_19_, *b*_21_–*b*_24_, *b*_27_, *y*_3_–*y*_11_, *y*_13_–*y_1_*_6_, *y_1_*_8_, *y_20_*–*y_27_* ions, combined with the key fragment ions (Table 2), indicated that its primary sequence was QQMCGRCGGTGQIIK_(Ac)_NECKTCHGKK_(Ac)_VTK, which corresponded with the Edman degradation data. This novel sequence was named Vi14b, since it was the second conotoxin discovered from *C. virgo* whose cysteine arrangement pattern (C-C-C-C) was framework XIV. Thereinto, two lysines (Lys, K) at the positions 15 and 25 of Vi14b were acylated since the mass addition of Lys was 42.0319, while the Lys at the positions 19, 24 and 28 were not modified. Acylation of lysine is a common PTM (posttranslational modification) among conotoxins.

Conotoxins Vi002 and Vi003 both contained no disulfide bridge since their molecular weights before and after TCEP reduction were the same. In MALDI–TOF–MS/MS spectrum of Vi002 (Figure 2), the consecutive ions *b*_1_–*b_3_*, *a*_1_–*a*_9_, *y*_13_–*y*_15_ and the crucial ions (*b*_10_, *b*_14_, *b*_17_, *b*_21_, *y*_1_, *y*_3_, *y*_5_, *a*_17_, *a*_20_, *a*_22_, *a*_24_), integrating with the key fragment ions (Table 3), explicitly revealed a novel sequence of LSSGATALSGVPRLTKPAGRLTTTTVAVAF. For sequence identification of Vi003, the serial *b*_1_–*b_7_*, *b*_9_–*b_11_*, *y*_3_–*y_4_*, *y*_7_–*y_9_*, *y*_11_ ions and the key fragment ions (KG m/z 186.1237, KGE m/z 287.1714, KGES m/z 402.1983, KGESL m/z 515.2824, KEGSLL m/z 628.3665, KEGSLLG m/z 685.3879) were observed in the tandem mass spectrum (Appendix A), indicating a novel sequence of NTESTKGESLLGK. It was a pity that we did not obtain enough Vi002 and Vi003 for Edman experiments. The amino acid “L” in the sequences of Vi002 and Vi003 represented “I/L”, since MS/MS sequencing could not distinguish leucine (L) and isoleucine (I). Herein, primary sequences of Vi002 and Vi003 were temporarily determined to be (I/L)SSGATA(I/L)SGVPR(I/L)TKPAGR(I/L)TTTTVAVAF and NTESTKGES(I/L)(I/L)GK, respectively.

Vi15a contained four pairs of disulfide bonds since the mass addition after the TCEP reduction was 8 Da (Appendix A). The MS/MS spectrum (Appendix A) showed successive ions *b*_1_–*b_8_*, *b*_13_–*b_16_*, *b*_18_–*b_22_*, *b*_24_–*b_26_*, *a*_22_–*a_26_*, *y_15_*–*y*_26_, *y_1_*–*y*_3_ and the key fragment ions (TCP *m*/*z* 302.1169, TCPW *m*/*z* 488.1962, TCPWG m/z 545.2177, DNCS *m*/*z* 420.1184, DNCSC *m*/*z* 523.1275, DNCSCI *m*/*z* 636.2116), which precisely determined the sequence to be DCTTCAGEECCGRCTCPWGDNCSCTEW-(nh_2_), a previously found disulfide-rich conotoxin Vi15a from *C. virgo* [24].

The mass differences between the reduced and unreduced conopeptides Mr3d.1, Mr3e.1, Mr3.8, Tx3a.1 and TCP were 6 Da (Appendix A), respectively, indicating that these five conotoxins all contained three pairs of disulfide bridges. The secondary MS spectra of the five conopeptides were shown in Appendix A, while their Edman degradation results were exhibited in Appendix A. The sequential ions *b*_2_–*b*_13_ observed in the MS/MS spectrum (Appendix A), combined with the deduced sequence of **RLS*GL**HP** from the Edman degradation results (Appendix A), ascertained a novel sequence of CCRLSCGLGCH**P**CC named as Mr3d.1. The Proline at the position 12 of Mr3d.1 was unmodified, while the 12-Proline of a known conotoxin Mr3d (CCRLSCGLGCH**O**CC-nh_2_, **O** presents “4-hydroxyproline”) [26,28] was hydroxylated and its C-terminal was amidated. Mr3e.1 was identified to be VCCPFGGCHELCYCCD since the consecutive ions *b*_2_–*b*_5_, *b*_7_–*b_1_*_5_ were detected in MS/MS spectrum (Appendix A), and the Edman degradation result (Appendix A) showed a sequence of V**PFGG*HEL*Y***. Herein, Mr3e.1 was a C-terminal deamidated variant of conotoxin Mr3e (VCCPFGGCHELCYCCD-nh_2_, [25,26,29]). Mr3.8 (CCHWNWCDHLCSCCGS) was a known conotoxin elucidated by serial ions *b*_3_–*b_15_*, *y_8_*–*y*_14_ in MS/MS data (Appendix A) and the Edman degradation experiment (**HWNW*DHL*S****, Appendix A). In Appendix A, the serial *y*_7_–*y*_9_, *y*_11_–*y*_15_, *b*_5_–*b*_6_ and *b*_8_–*b*_15_ ions clearly confirmed the sequence to be CCSWDVCDHPSCTCCG, which was a C-terminal carboxylated variant of a known conotoxin Tx3a (CCSWDVCDHPSCTCCG-nh_2_) from *C. texile* [27,28]. Thus, it was named Tx3a.1. The sequence for Appendix A was assigned NCPYCVVYCCPPAYCEASGCRPP (a previously found conotoxin named TCP [27]) by successive *b*/*y* ions (*y_1_*–*y*_22_, *b*_1_–*b_3_*, *b*_5_–*b*_9_, *b*_11_, *b*_21_–*b*_22_). The Edman degradation result (N*PY*****VVY*******PAY*****E*SG********, Appendix A) also supported the above sequence deduced from MS/MS sequencing.

The existence of disulfide bonds is of great significance for stabilizing conotoxins’ spatial structures, enhancing activity and improving selectivity to target receptors [30]. Different disulfide linkages (quantity or connectivity pattern) produce different three-dimensional structures, which lead to differences in activity and selectivity afterwards [30,31,32]. In this work, we isolated a novel two-disulfide-bonded conotoxin Vi14b, two novel disulfide-poor conotoxins Vi002 and Vi003, and five three-disulfide-bridged conotoxins Mr3d.1, Mr3e.1, Mr3.8, Tx3a.1 and TCP. Conotoxins Mr3.8 and TCP were previously discovered from *Conus* venom isolation or transcriptomic/proteomic analyses [25,26,27,28,33]. C.D. Poulter et al. synthesized Tx3a and characterized its disulfide connectivity as I-V, II-IV, III-VI. They found that Tx3a can lead to an excitation response by injecting it into mice [34]. TCP (textile convulsant peptide) was an *O*-superfamily convulsant peptide with a VI/VII Cys framework purified from *C. texitle* venom components that had induced dramatic and diverse symptoms in mice [28,35,36]. To date, no other record of biological activity and target for Mr3.8, Tx3a.1 and TCP has been reported.

Since the amount of the isolated peptides in this work was quite limited, in order to investigate the structure and activity of these disulfide-rich conotoxins, we synthesized conotoxins Vi14b, Mr3d.1, Mr3e.1, Mr3.8, Tx3a.1 and TCP using a one-step oxidation strategy. For every conotoxin, we individually performed the HPLC co-elution of the isolated peptide and its oxidative products in order to track the desired isomer that was consistent with the native peptide. Subsequently, the correct isomer was purified by HPLC system. The homogeneities between the synthetic peptide and the corresponding isolated one were verified by the HPLC co-elution experiment.

### 2.2. Disulfide Connectivity

Conotoxins Vi14b, Mr3d.1, Mr3e.1, Mr3.8, Tx3a.1 and TCP were synthesized using a one-step oxidation strategy. The disulfide connectivities of these folded peptides were unknown. Herein, their disulfide linkages were determined by MALDI–TOF–MS/MS analysis of their two-cysteine-alkylated products (product I) and the four-cysteine-alkylated products (product II) generated from the stepwise reduction and subsequent alkylation of these folded peptides (Figure 3 and Appendix A).

The MS/MS spectrum of the reduced two-cysteine-alkylated product (product I, Figure 3) of Vi14b was shown in Figure 4. We used “C_(IAM)_” to represent the cysteine that was alkylated by IAM (Iodoacetamide) in Figure 4, Figure 5 and Figure 6 and Appendix A. The mass differences of *y*_25_/*y*_24_ and *y_8_*/*y_7_* were 160.0307 and 160.0306, respectively, while mass differences of *y*_22_/*y*_21_ and *y*_11_/*y*_10_ were both 103.0091, indicating that Cys-I and Cys-IV were alkylated while Cys-II and Cys-III were in their reduced form. It meant that a disulfide linkage existed between Cys-I and Cys-IV. Herein, the disulfide connectivity of Vi14b was characterized as I-IV, II-III.

For product I (Figure 3) of Mr3d.1, Cys at positions 2 and 10 (2-Cys and 10-Cys) were speculated to be IAM-labeled since the mass difference (160.0307) of *y*_13_/*y*_12_ and *b_10_*/*b*_9_ (or *y_5_*/*y_4_*) was deduced from its MS/MS spectrum (Figure 5). In the MS/MS spectrum of product II of Mr3d.1 (Figure 6), the mass difference (160.0307) of *y*_13_/*y*_12_, *y*_9_/*y*_8_, *y*_5_/*y*_4_ (or *b*_10_/*b*_9_) meant the Cys at positions 2, 7 and 10 were alkylated. The observation of ions *y*_14_/*y*_13_ and fragment ions C_(IAM)_HPC (*m*/*z* 498.1588) and C_(IAM)_HPCC_(IAM)_ (*m*/*z* 573.1731) suggested that thiols in 1-Cys and 13-Cys were in reducing state and 14-Cys were alkylated. Herein, the disulfide linkage of Mr3d.1 was confirmed to be I-V, II-IV, III-VI.

As for product I of Mr3e.1 (Appendix A), 3-Cys and 14-Cys were ethyl-labeled because of the mass difference (160.0307) of *b*_3_/*b*_2_ (or *y*_14_/*y*_13_) and *b*_14_/*b*_13_, while the other four cysteines were dissociative. In MS/MS spectrum of product II of Mr3e.1 (Appendix A), the ions of *y*_15_/*y*_15_, *b*_3_/*b*_2_, *b*_8_/*b*_7_, *b*_14_/*b*_13_ revealed the alkylation of Cys at positions 2, 3, 8 and 14. Thus, Mr3e.1 was considered to have I-III, II-V, IV-VI disulfide connectivity. Similarly, for Mr3.8, Cys-II and Cys-VI of its product I (Appendix A) were alkylated by the observation of *y*_15_/*y*_14_ and *b*_14_/*b*_13_. In MS/MS spectrum of product II of Mr3.8 (Appendix A), the detection of ions *b*_14_ (*m*/*z* 458.1275) and *y*_3_ (*m*/*z* 323.1020) indicated that the cysteins at positions 1, 2 and 14 were IAM-labeled, while 2-Cys and 10-Cys were in the reducing state due to the mass difference (103.009) of *b*_7_/*b*_6_ and *b_11_*/*b_10_*. The 13-Cys were alkylated since *y*_a_ ion (*m*/*z* 573.2272) of the fragment HLCSC_(IAM)_ was detected. Thus, disulfide linkage of Mr3.8 was identified as I-V, II-VI, III-IV.

With regard to Tx3a.1, Cys-I and Cys-III were ascertained to be alkyl-marked in its product I due to the mass difference (160.03) of ions of *y*_10_/*y*_9_ and *y*_15_/parent ion (*m*/*z* 1832.5632) (Appendix A), while ions of *y*_15_/*y*_14_, *b*_12_/*b*_11_, *b*_14_/*b*_13_, *b*_15_/*b*_14_ revealed Cys at positions 2, 12, 14 and 15 were in their reduced form. For product II of Tx3a.1 (Appendix A), the ions *b*_2_ (*m*/*z* 321.0608), *y*_2_ (*m*/*z* 236.0700) and *y*_10_/*y*_9_ (or *b*_7_/*b*_6_) ascertained 1-, 2-, 7-, 15-Cys were alkyl-labeled. Thus, the disulfide connectivity of Tx3a.1 was assigned as I-III, II-VI, IV-V. Similarly, disulfide linkages of TCP were notarized as I-III, II-V, IV-VI because of ions of *b*_12_/*b*_11_ and *y*_15_/*y*_14_ were observed in the MS/MS spectrum of its product I (Appendix A) and ions of *b*_2_ (*m*/*z* 275.0809), *y*_19_/*y*_18_, *y*_15_/*y*_14_, *y*_9_/*y*_8_ were detected in MS/MS spectrum of its product II (Appendix A).

Thus, the disulfide connectivities of conotoxins Vi14b, Mr3d.1, Mr3e.1, Mr3.8, Tx3a.1 and TCP were assigned as shown in Figure 7. Generally, linkage patterns of I-V, II-IV, III-VI and I-III, II-V, IV-VI are common among the conotoxins with three pairs of disulfide bonds, while the patterns of I-V, II-VI, III-IV (such as Mr3.8) and I-III, II-VI, IV-V (such as Tx3a.1) were rarely reported and different from that previously established for the µ- and ψ-conotoxins [37]. The novel connectivity patterns could possibly cause these kinds of conotoxins, such as Tx3a.1 and Mr3.8, to present specific biological activity.

### 2.3. Blockage on Voltage-Gated Sodium Channel

The Cys arrangement pattern of Mr3d.1, Mr3e.1, Mr3.8, Tx3a.1 was framework III, while that of TCP was framework VI/VII. Frameworks III and VI/VII are common in μ-conotoxins, which generally target voltage-gated sodium channels (VGSC). Sodium channel subtypes can be classified as TTX-sensitive (Na_v_1.1, 1.2, 1.3, 1.4, 1.6 and 1.7), or TTX-resistant (Na_v_1.5, 1.8, 1.9) channels upon their sensitivity to tetrodotoxin (TTX) [13]. Na_v_1.4 is highly expressed in adult skeletal muscle and is associated with disorders such as myotonia, myasthenia and periodic paralysis [23]. A potent and selective Na_v_1.4 blocker μ-CnIIIC (IC_50_ 1.3 nM, framework III) has been commercialized as an instant line relaxer called XEPTM-018 and been used in a facial cream (Lirikos^TM^, Amorepacific Corporation) for wrinkle smoothing [38]. Na_v_1.8 is generally distributed in DRG (dorsal root ganglion) neurons and is considered to be an important target for chronic pain therapy [39]. μO-MrVIB (IC_50_ 326 nM for Na_v_1.8, framework VI/VII) is a potential drug for neuropathic pain treatment, and its investigation has entered clinical phase II [40]. Here, we evaluated the effect of conotoxins Mr3d.1, Mr3e.1, Mr3.8, Tx3a.1, TCP on Na_v_1.4 and Na_v_1.8 channels. The result (Table 4 and Figure 8) indicated that conotoxins Mr3e.1, Mr3.8, Tx3a.1, and TCP blocked Na_v_1.4 or Na_v_1.8 by slight inhibition on the current at the complete resting state (TP1), while they inhibited Na_v_1.4 or Na_v_1.8 by 21.51~24.32% of currents in a semi-activated state (TP2) at 10 μmol/L. Other tested conotoxins showed no current inhibition on Na_v_1.4 or Na_v_1.8. Activities on other subtypes of VGSC or other types of ion channels remain to be determined.

### 2.4. Tx3a.1, Mr3.8 and Vi14b Inhibited the Proliferation of ID8 Cells

To study the anticancer potential of conotoxins Vi14b, Mr3d.1, Mr3e.1, Mr3.8, Tx3a.1, TCP, the cytotoxicity of these peptides against ID8 ovarian cancer cells was detected by MTT assay. We treated the cells with different concentrations (0, 5, 10, 20, 40, 80, 160 μM) of peptides for 72 h. These peptides showed significant inhibitory effect on ID8 cells at a low concentration of 5 μM compared with the control group, especially Tx3a.1. The cell inhibitory rates were 45.08%, 23.79% and 19.39% for Tx3a.1, Mr3.8 and Vi14b, respectively, at a concentration of 10 μM. A stronger effect was observed after exposure to higher concentrations of Tx3a.1, with a 50% reduction in cell viability at a concentration of 24.29 μM. Mr3.8 killed half the cells at a concentration of 54.97 μM. The IC_50_ of Vi14b was 111.6 μM. The data (Figure 9) were presented as mean ± SD of three independent experiments. These results indicated that the Tx3a.1, Mr3.8 and Vi14b exhibited strong anticancer effects on ID8 cells in vitro. It was the first report of conotoxins showing anti-ovarian cancer activity. The mechanism of their anti-ovarian cancer activity remains to be explored later. In this work, we identified and structurally and biologically characterized the anti-ovarian cancer conotoxins from *Conus* venom. The result delivered a message that conotoxins, a class of active polypeptides with diverse structures, could be a rich source of anti-ovarian cancer drug candidates. Herein, this study highlights the role of conotoxin library in subsequent drug discovery for ovarian cancer treatment.

## 3. Materials and Methods

### 3.1. Venom Preparation and Peptide Isolation

Cone snail specimens were collected from coastal waters near Sansha city in the South China Sea and were frozen at −80 °C for hours. Dr. Ying Fu identified the specimens. The venom duct of the snails was dissected, and the venom duct cut into small sections (about 1cm long). They were extracted with 60% acetonitrile aqueous solution at 4 °C for 24 h, then centrifuged at 8000 r/min for 10 min. The supernatant was pipetted and filtered with 0.22 µm organic phase pinhole filter, then 60% acetonitrile aqueous solution was added to the precipitate for secondary extraction. This was repeated twice and the filtrate combined to obtain the crude venom solution, which was subjected to lyophilization to yield crude venom powder.

The venom powder was dissolved in 3% acetonitrile aqueous solution and filtrated. The venom solution was fractionated using a preparative high-performance liquid chromatography (Pre-HPLC) instrument (Waters, Milford, MA, USA) with a C_18_ column (Vydac Grace, 10 µm, 22 mm × 250 mm). The preparative system was conducted by linear gradient elution with 5 to 90% solution B (90% acetonitrile aqueous solution with 0.1% TFA) for 60 min at an 8 mL/min flow rate, while solution A was double-distilled water (ddH_2_O) with 0.1% TFA. The fractionated components were lyophilized and redissolved in 10% acetonitrile aqueous solution, then were subjected to a HPLC system equipped with a C_18_ column (Vydac Grace, 5 µm, 4.6 mm × 250 mm). The separation process was performed with mixed eluant (solution A and B) at a flow rate of 0.8 mL/min. Conopeptides were yielded from the venom fractions by a 20-min linear gradient program of 30% to 50% solution B (namely 70% to 50% solution A) detecting by ultraviolet (UV) absorption at 214 nm.

The molecular weights of the purified conopeptides were determined using an ultra-performance liquid chromatography–triple quadrupole-mass spectrometry (UPLC–TQD–MS, Waters, USA) spectrometer with a BEH C_18_ column (130 A, 1.7 µm, 2.1 mm × 100 mm). Conditions were set as follows: detection range *m*/*z* 400–1500, cone voltage 30 V, capillary voltage 3.5 kV, dissolution flow rate 1000 L/h, dissolution temperature 500 °C, ion source temperature 150 °C. The UPLC program was performed by gradient elution from 5 to 60% solution B for 6 min at a flow rate of 0.5 mL/min.

### 3.2. Peptide Sequencing

The isolated peptides were individually alkylated by iodoacetamide to confirm whether they contained any free thiol or not. In order to ascertain the number of disulfide bonds in these conopeptides, we totally reduced the peptides by TCEP and detected their mass differences between intact peptides and the reduced forms (Figure 3), respectively.

Then the purified conotoxins were totally reduced to obtain linear peptides before sequencing. Each peptide was dissolved in 20 µL 50% acetonitrile aqueous solution. Then, 10 μL 50 mmol/L TCEP and 10 μL NH_4_HCO_3_ were added to the peptide solution, which was incubated at 25 °C for 30 min. The reaction was monitored by UPLC analysis at 214 nm. The molecular weight of each reduced peptide was determined by UPLC–TQD–MS over *m*/*z* 100–2000.

Sequence identification was achieved by MALDI–TOF (matrix-assisted laser desorption—ionization—time of flight) tandem mass spectrometer (Bruker, Ultraflextreme, Germany). The reduced linear peptides were dissolved in 50% acetonitrile aqueous solution. The matrix solution was prepared by dissolving 1 mg cyano-4-hydroxycinnamic acid (HCCA) in 100 µL standard solution (47.5 µL ddH_2_O, 50 µL acetonitrile, 2.5 µL trifluoroacetic acid). For sample loading, 1 µL of each reduced peptide solution was individually dropped onto different circles on the plate and then dried off. One µL of matrix solution was successively dropped onto the same position on the plate. The sample plate was then loaded into the spectrometer when the solution was completely dried off. FlexControl was used to acquire MS data, while FlexAnalysis was applied to mark the masses. Conditions were set as follows: laser frequency 1000 Hz, laser energy range 20–40%, ion source voltage 7.5 kV. The voltages for secondary mass detection were 19.00 kV for Lift 1 and 2.80 kV for Lift 2. For the sequence identification, ion masses and related fragmentation masses in the secondary mass spectrum of each conopeptide were matched by Mascot search (http://www.matrixscience.com/cgi/nph-mascot.exe?1, accessed from 1 December 2020 to 23 February 2022) against the NCBI and Swiss-prot protein database. Modifications involved in the search were as follows: amidation, deamidation, hydroxylation of proline and valine, oxidation of methionine, carboxylation of glutamic acid, cyclization of N-terminal glutamine (pyroglutamate), bromination of tryptophan, and sulfation of tyrosine.

An Edman degradation experiment was performed for sequence elucidation as well. The solution was vortexed until the reduced peptide powder was dissolved in 50 μL ddH_2_O. Then, 10 μL of the peptide solution was dropped onto the polyvinylidene fluoride (PVDF) membrane and dried off. The PVDF membrane was transferred onto a PPSQ-53A protein sequencer (Shimadzu, Japan). Each phenylthiohydantoin (PTH)-amino acid generated from every cycle of N-terminal degradation of the peptide was analyzed by HPLC system equipped with a Wakosil PTH-II column (Wako, 4.6 mm × 250 mm). For the mobile phase, 40% acetonitrile aqueous solution with less than 2.5% acetic acid was used. The column temperature was set to 40 °C, and the flow rate was 1.0 mL/min. The retention time of each (PTH)-amino acid was compared with that of the corresponding standard amino acid to identify the primary amino acid sequence of the isolated conopeptides.

### 3.3. Peptide Synthesis

The linear conopeptides were separately synthesized using Fmoc-SPPS (9-Fluorenylmethoxycarbonyl-peptide solid phase synthesis) chemistry [41]. The Fmoc-amino acids were purchased from Chengdu Chron Chemical Co., LTD. Amino acid ligation was operated on the Wang-resin with HBTU/DIEA used as the coupling reagent. Kaiser test regent was applied to detect the state of every coupling cycle. Piperidine/DMF (*v*/*v* 20:80) was used to remove the Fmoc groups. The cysteine residues were protected by triphenylmethyl (trt) groups, which were then removed along with the peptide release process. Cleavage cocktail (TFA/thioanisole/phenol/EDT/H_2_O 87.5:2.5:2.5:2.5) was used to release the linear peptide from the resin at room temperature for 2 h, filtrate to remove the resin, and the filtrate was treated with cold ether (0 °C) overnight to precipitate the crude peptide. The precipitation was washed with cold ether twice and lyophilized to obtain crude peptide power, which was subjected to Pre-HPLC separation to yield pure linear peptide.

The obtained linear conopeptide was subjected to one-step air oxidation in a 0.1 M Tris-HCl/1 mM EDTA buffer with GSH/GSSG (1 mmol/L/0.5 mmol/L) at 40 °C for 60 min. The linear peptide concentration ranged from 50 μM to 100 μM. The oxidative products were detected by LC–MS and subsequently purified by HPLC system with a 20-min linear gradient elution of 30% to 50% solution B at a 0.8 mL/min flow rate.

### 3.4. Disulfide Connectivity Recognition

The stepwise reduction and alkylation protocol were performed as reported [42]. The folded peptide (20–30 μg in 100 μL ddH_2_O containing 0.1% TFA) was partially reduced by mixing with an appropriate volume of 100 mM TCEP dissolved in citrate buffer (0.1 M, pH 3.0) to a final concentration of 6 mM TCEP. The mixture was incubated at 40 °C ranging from 5 min to 25 min for folded peptide to generate the corresponding one-disulfide-reduced product and two-disulfide-reduced product. The reaction state was monitored by LC–MS to confirm that the exact partially reduced product was generated. Then the reaction mixture was blended with alkylation solution (9.24 mg iodoacetamide dissolved in 25 μL acetonitrile and 75 μL 1 mM pH 8.0 Tris-HCl buffer) to alkylate the free thiol groups of the partial-reduced products. The alkylation reaction was conducted at 40 °C for 15–45 min and monitored by LC-MS. Then the mixture was subjected to going through HPLC column (Vydac Grace, 5 μm, 4.6 mm × 250 mm) to terminate the reaction and purify the alkyl-labeled products. The obtained two-cysteine-alkylated and four-cysteine-alkylated products were subsequently reduced by TCEP to break the remaining disulfide bonds to gain linear peptides, which were named as products I and II, respectively. Then the products I and II of the synthetic conopeptide were subjected to MALDI–TOF–MS/MS analysis to confirm which thiol groups were alkylated.

### 3.5. Patch Clamp Recording for Voltage-Gated Sodium Channel

Blocking activity of voltage-gated sodium channels was conducted using the whole-cell patch clamp technique [43]. CHO cells were cultured in HAM’S/F-12 medium containing 10% fetal bovine serum. 6.5 × 10^3^ cells were plated on coverslips, cultured in 24-well plates (final volume 500 µL). Patch clamp assays were performed after 18 h. Solution preparation involved intracellular fluid Na_v_-001-2 (50 mM CsCl, 10 mM NaCl, 10 mM HEPES, 60 mM CsF, 20 mM EGTA, adjusted to pH 7.2 with CsOH), extracellular fluid K-007-1 (140 mM NaCl, 3.5 mM KCl, 1 mM MgCl_2_•6H_2_O, 2 mM CaCl_2_•2H_2_O, 10 mM *D*-Glucose, 10 mM HEPES, 1.25 mM NaH_2_PO_4_•2H_2_O, adjusted to pH 7.4 with NaOH). The testing peptide was dissolved with DMSO to a concentration of 10 mM solution. The peptide solution was diluted to 10 µM using extracellular fluid. Tetrodotoxin (TTX, 1 µM) dissolved in ddH_2_O containing 0.1% acetic acid was used as a Na_v_1.4 positive control, while Na_v_1.8 positive control was prepared by dissolving 10 mg A-803467 [5-(4-chlorophenyl-N-(3, 5-dimethoxyphenyl) furan-2-carboxamide] with DMSO and diluting it to 10 µM by K-007-1). The microelectrode manipulator was operated under an inverted microscope (MP285, Sutter Instrument, Novato, CA, USA). The recording electrode was placed on the cell, and negative pressure was applied to form a GΩ seal. Then, fast capacitance compensation was performed and negative pressure applied until the cell membrane was sucked to form a whole-cell recording mode. The membrane voltage was clamped at −120 mV after whole-cell sealing. Cell-plated coverslips were placed in the recording bath in an inverted microscope. The testing peptide solution and the peptide-free extracellular fluid flowed through the recording bath from low to high concentration by gravity perfusion. Each peptide was reacted for 5 min before the next peptide was detected. The data were collected by EPC-10 amplifier (HEKA, Reutlingen, Baden Wurdenburg, Germany) and stored by PatchMaster software (HEKA, Reutlingen, Baden Wurdenburg, Germany).

The reaction rate on voltage-gated sodium channel were defined as $\frac{Peak\;current_{conotoxin}}{Peak\;current_{control}}$, and the percentage of inhibition was determined as 1 − $\frac{Peak\;current_{conotoxin}}{Peak\;current_{control}}$ × 100%. Meanwhile, the mean and standard deviation were calculated through three parallel experiments. The voltage-gated sodium channel activity data was analyzed by PatchMaster (HEKA, Reutlingen, Germany) & IGOR Pro (Wave Metrics, Portland, OR, USA). In addition, data processing and graphing were performed by the software GraphPad Prism 6.0 (GraphPad Software, San Diego, CA, USA).

### 3.6. Ovarian Cancer Cell Viability Assay

The MTT method was used to determine the cytotoxicity of the synthetic conotoxins [44]. Briefly, ID8 cells (4 × 10^3^ cells/well) were seeded in 96-well plates. After 12 h incubation, the culture medium was removed and the cells were treated with different concentrations of peptides for 72 h. After adding 1 mg/mL MTT solution, the cells were cultured for another 4 h. Subsequently, 100 µL DMSO was added to dissolve the formazan. The absorbance was measured at 570 nm using a SpectraMax^®^ M2 microplate reader (Molecular Devices, Sunnyvale, CA, USA). The IC_50_ values were expressed as mean ± SD of individual samples and were representative of three independent experiments. The data were analyzed by unpaired two-tail *t*-test (Student’s *t*-test). Values of *P* less than 0.05 were regarded as statistically significant. Calculation of IC_50_ values and statistical analysis were performed using GraphPad Prism 6.0 (GraphPad Software, San Diego, CA, USA).

## 4. Conclusions

We isolated and identified three novel conopeptides named as Vi14b, Vi002, Vi003, three conotoxin variants named as Mr3d.1, Mr3e.1, Tx3a.1, and three known conotoxins (Vi15a, Mr3.8 and TCP) from *Conus* venom. Mr3.8 and Tx3a.1 both showed a novel pattern of disulfide connectivity, different from that previously established in the µ- and ψ-conotoxins. Tx3a.1, Mr3e.1 and Vi14b exhibited a certain anti-ovarian cancer effect on ID-8 cells in vitro. It was the first report of conotoxins showing anti-ovarian cancer activity. This work highlights the role of conotoxin library in subsequent drug discovery for ovarian cancer treatment.

## Figures and Tables

**Figure 1 molecules-27-06609-f001:**
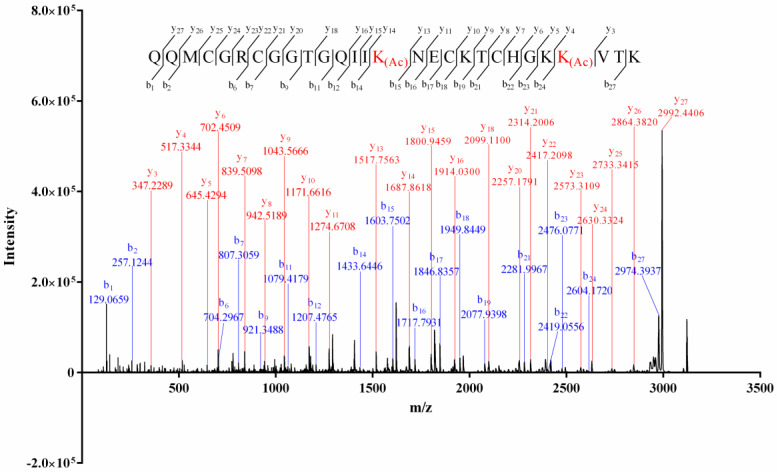
Sequencing of conotoxin Vi14b by MALDI–TOF–MS/MS analysis.

**Figure 2 molecules-27-06609-f002:**
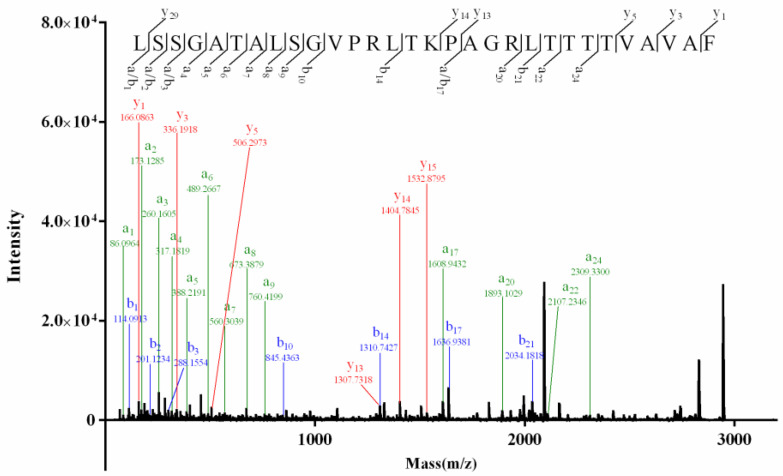
MALDI–TOF–MS/MS spectrum of conotoxin Vi002.

**Figure 3 molecules-27-06609-f003:**
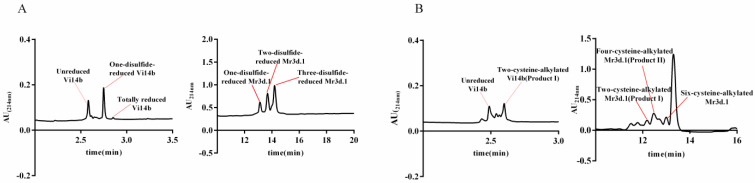
HPLC analysis of stepwise reduction by TCEP and follow-up alkylation by IAM. (**A**) Stepwise reduction products of Vi14b and Mr3d.1. (**B**) Alkylated products of stepwise-reduced Vi14b and Mr3d.1.

**Figure 4 molecules-27-06609-f004:**
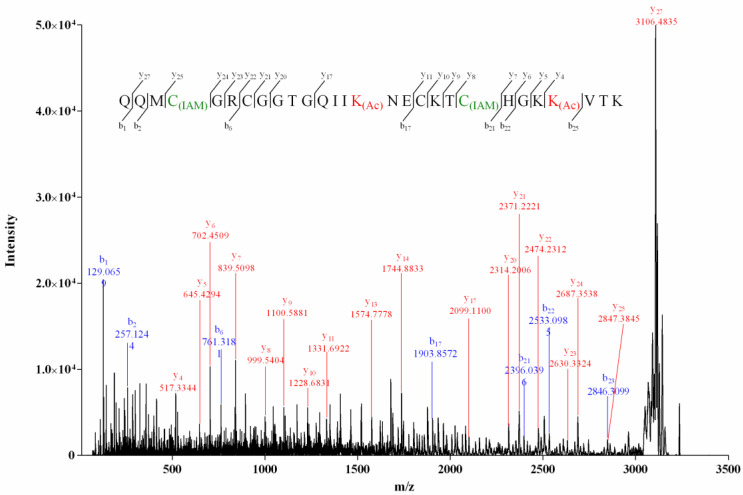
MALDI–TOF–MS/MS spectrum of product I of Vi14b.

**Figure 5 molecules-27-06609-f005:**
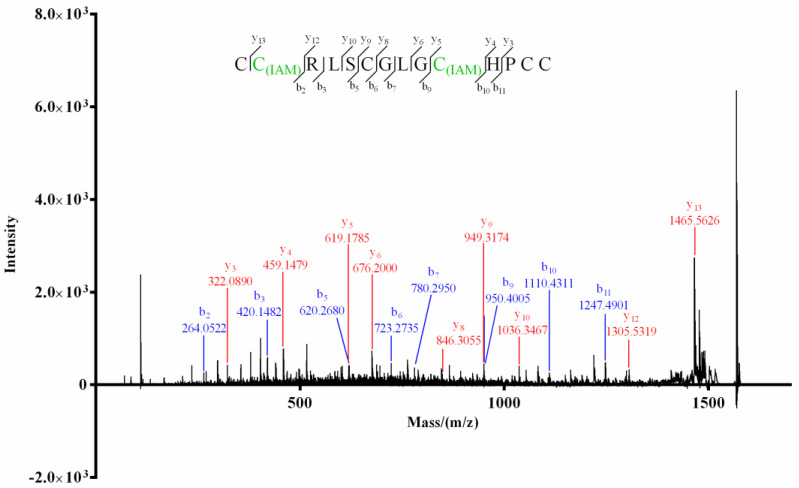
MALDI–TOF–MS/MS spectrum of product I of Mr3d.1.

**Figure 6 molecules-27-06609-f006:**
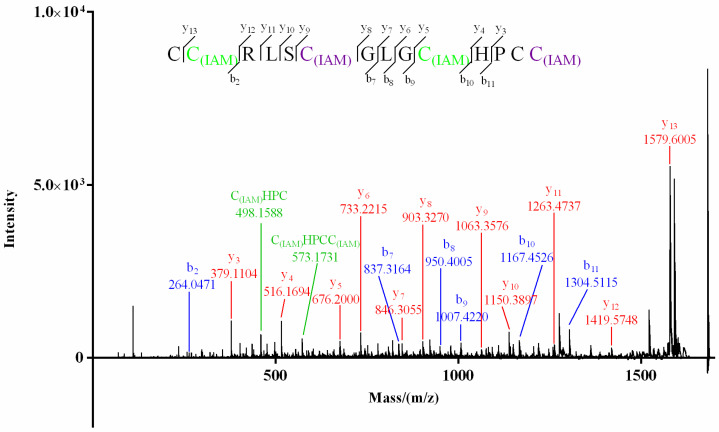
MALDI–TOF–MS/MS spectrum of product II of Mr3d.1.

**Figure 7 molecules-27-06609-f007:**
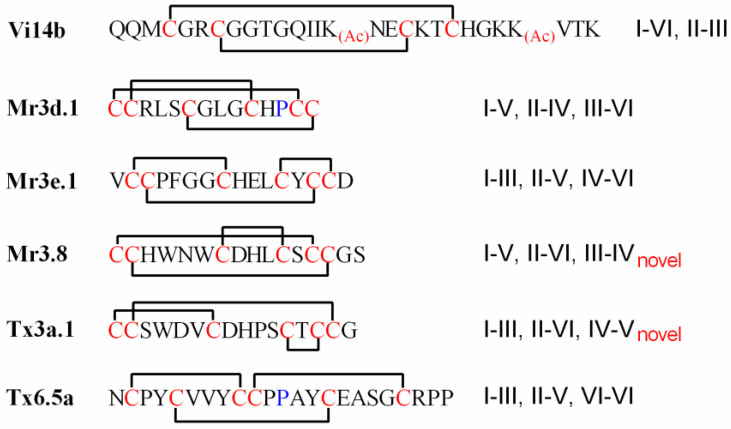
Disulfide connectivities of Vi14b, Mr3d.1, Mr3e.1, Mr3.8, Tx3a.1 and TCP.

**Figure 8 molecules-27-06609-f008:**
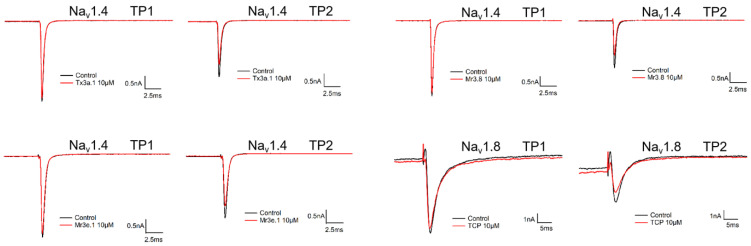
Blocking effects of Tx3a.1, Mr3.8, Mr3e.1 and TCP on Na_v_1.4 and Na_v_1.8.

**Figure 9 molecules-27-06609-f009:**
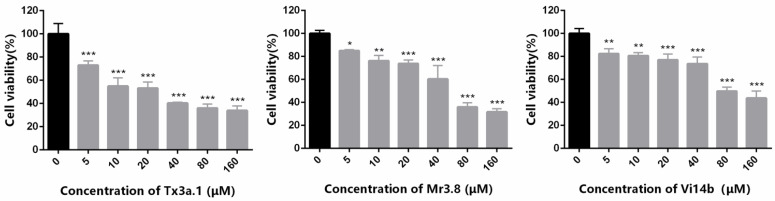
Ovarian cancer cytotoxicity of Tx3a.1, Mr3.8 and Vi14b in ID8 cells (Student’s *t*-test, * *p* < 0.05; ** *p* < 0.01, *** *p* < 0.001).

**Table 1 molecules-27-06609-t001:** Conotoxins identified from *Conus* venoms.

Name	Sequence	Origin	Disulfide Bridge	Reference
Vi14b	QQMCGRCGGTGQIIK_(Ac)_NECKTCHGKK_(Ac)_ VTK	*C. virgo*	2 pairs	This work
Vi002	LSSGATALSGVPRLTKPAGRLTTTTVAVAF *	*C. virgo*	none	This work
Vi003	NTESTKGESLLGK *	*C. virgo*	none	This work
Vi15a	DCTTCAGEECCGRCTCPWGDNCSCTEW-(nh_2_)	*C. virgo*	4 pairs	[24]
Mr3d.1	CCRLSCGLGCHPCC	*C. marmoreus*	3 pairs	This work
Mr3e.1	VCCPFGGCHELCYCCD	*C. marmoreus*	3 pairs	This work
Mr3.8	CCHWNWCDHLCSCCGS	*C. marmoreus*	3 pairs	[25,26]
Tx3a.1	CCSWDVCDHPSCTCCG	*C. texile*	3 pairs	This work
TCP	NCPYCVVYCCPPAYCEASGCRPP	*C. texile*	3 pairs	[27]

* The amino acid “L” in the sequence represented “I/L”, since MS/MS sequencing could not distinguish leucine (L) and isoleucine (I).

**Table 2 molecules-27-06609-t002:** Mass of key fragment ions in MS/MS spectrum of conotoxin Vi14b.

Sequence	y_a_/y_b_	Sequence	y_a_/y_b_	Sequence	y_a_/y_b_
RCGGT	475.2082	TGQIIK_(Ac)_	683.4087	GKK_(Ac)_	356.2292
RCGGTG	504.2347	IIK_(Ac)_	397.2809	GKK_(Ac)_V	427.3027
CGGTGQ	504.1871	IK_(Ac)_N	398.2398	GKK_(Ac)_VT	528.3504
CGGTGQI	617.2712	IK_(Ac)_NE	527.2824	KK_(Ac)_V	398.2762
TGQI	372.2241	HGK	323.1826	K_(Ac)_VT	371.2289
IK_(Ac)_	284.1969	GK	186.1237		

* y_b_ ion represented the y ion of the intact sequence fragment, while y_a_ was the fragment ion whose C-terminal amino acid residue was decarboxylated.

**Table 3 molecules-27-06609-t003:** Mass of key fragment ions in MS/MS spectrum of conotoxin Vi002.

Sequence	y_a/_y_b_	Sequence	y_a/_y_b_	Sequence	y_a/_y_b_
SSG	232.0928	GVP	226.1550	LTT	316.1867
SSGA	303.1299	PR	226.1662	AGRLTTT	673.3992
SSGAT	404.1776	LT	187.1441	RLTTTT	674.3832
GATA	301.1506	LTK	315.2391	TTTV	403.2187
GATAL	386.2398	TKPAG	455.2613	TTTVA	474.2558
GATALS	473.2718	PAG	226.1186	TTTTVAV	545.3293
GATALSG	558.2882	AGR	285.1670	TTVAV	472.2766
SGVP	341.1819	GRL	299.2190	VAVA	341.2183
GV	129.1022	LT	187.1441		

**Table 4 molecules-27-06609-t004:** Current inhibition of the tested conotoxins on Na_v_1.4 and Na_v_1.8.

Name	TP1	TP2
Mr3e.1	3.79 ± 0.61% on Na_v_1.4	21.89 ± 3.06% on Na_v_1.4
Mr3.8	4.62 ± 1.06% on Na_v_1.4	23.61 ± 2.70% on Na_v_1.4
Tx3a.1	3.17 ± 0.63% on Na_v_1.4	24.32 ± 2.29% on Na_v_1.4
TCP	5.45 ± 0.39% on Na_v_1.8	21.51 ± 1.00% on Na_v_1.8

## Data Availability

Data is contained within the article or Appendix A.

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
