# Peer review of "Anti-Ovarian Cancer Conotoxins Identified from *Conus* Venom"

_molecules, 2022, doi:10.3390/molecules27196609_

Round 1
Reviewer 1 Report
This study presents the de novo sequencing, synthesis, disulfide connectivity recognition and biological evaluation of the isolated conotoxins from crude venoms of C. virgo, C. marmoreus and C. texile. Mr3.8 and Tx3a.1 both showed a novel pattern of disulfide connectivity, different from that previously established in the µ- and ψ-conotoxins.
I think the following issues need to be discussed in the manuscript:
(1) These peptides Mr3d.1, Mr3e.1, Mr3.8, Tx3a.1 and TCP have 6 cysteines to form 3 disulfide bonds. In this study, the one-step air oxidation method is used to synthesize peptides, which will form 8 isomers. Please explain how to obtain the correct and desired isomers.
(2) These peptides are not particularly good at blocking sodium ion channels. Have you screened other ion channel activities?
(3) I would like to ask how to judge whether these peptides have anti-ovarian cancer effects and whether they have been screened for other types of cancer cells.
Author Response
Dear Reviewer:
Thanks very much for your valuable comments.
Our answers to your comments are as followed:
- We synthesized all the peptides by one-step oxidation strategy by which the disulfide bridges were randomly formed. Certainly, several isomers were generated. We performed the HPLC co-elution of the isolated peptide and its oxidative products in order to track the desired isomer that was consistent with the native peptide. Subsequently, the correct isomer was purified by HPLC system. These statements have been added to lines 169-173.
- Due to the limited experimental conditions, we only tested their inhibitory activities on two subtypes of voltage-gated sodium channels (VGSC), namely Nav4and Nav1.8. Activities on other subtypes of VGSC or other types of ion channels remains to be determined.
- Due to the limited conditions, we only determined their anti-ovarian cancer effect in vitro on the ovarian cancer cell line ID8. We judged their anti-ovarian cancer effects by the explicit inhibition on the proliferation of ID8 cells since the significant differences were shown between the peptide administration groups (at 5, 10, 20, 40, 80, 160 μM) and the control group (Figure 9).
Thanks again and wish you a good day.
Sincerely,
Dr. Ying Fu
Reviewer 2 Report
1 In Line 43 Conotoxins (Ctx) is a class of active polypeptides secreted by cone snails (a carnivo- 43 rous mollusk living in tropical oceans) and it is mainly used for prey capture and defense 44 against natural enemies [10,11]. The following reference could be added here
Yihe Zhao, Agostinho Antunes. Biomedical Potential of the Neglected Molluscivorous and Vermivorous Conus Species. Mar Drugs. 2022; 20(2): 105.
2 According to author guidelines, References should be described as follows Journal Articles:
1. Author 1, A.B.; Author 2, C.D. Title of the article. Abbreviated Journal Name Year, Volume, page range.
A) Please check in the item References numbers 25, 26 and 42
Last numbers of page ranges are incomplete
B) References 30 and 38
Please add USA after Proc Natl Acad Sci
C) In ref 35 year of publication is 2028
Author Response
Dear Reviewer:
Thanks very much for your valuable comments.
Our answers to your comments are as followed:
The reference has been added to the reference list. The style of the references has been revised based on the journal instructions.
Thanks again and wish you a good day.
Sincerely,
Dr. Ying Fu
Reviewer 3 Report
07 Sept -2022
Journal: Molecules
Title: Anti-Ovarian Cancer Conotoxins Identified from Conus Venom
Dear Editor:
The authors have evaluated the anticancer activity of Conotoxins derived from Conus Venom. The manuscript carries scientific merit and novelty and would be suitable for publication after major revision as suggested below.
Nermeen Yosri, PhD
Comments to authors:
1- The graphical abstract is highly recommended.
2- Please add the most relevant results to the abstract
3- " Conotoxins is a treasure of huge drug resource." Please add examples
4- " This work highlights the role of conotoxin library in subsequent drug discovery for ovarian cancer"; please explain in more detail
5- " alternative antitumor drugs are urgently needed for the treatment of ovarian cancer "; could you add some examples
6- " To date, more than 10 contoxins, such as ω-MVIIA, ω-CVID, contulakin-G, conantokin-G, χ-MrIA, α- Vc1.1, κ-PVIIA, μO-MrVIB, μ-SIIIA, CGX-1204, etc., have made it to the clinical research stage "; For which diseases and which phases?
7- Figures are not clear; please improve the quality
8- Please add a list of abbreviations
9- Who identifies the specimens?
10- " IC50" should be IC50
11- Could you add static analysis?
12- The conclusion section should be reduced by focusing on the main outcomes of the study.
13- The authors could benefit from the following reference:
Yosri, N., Khalifa, S.A., Guo, Z., Xu, B., Zou, X. and El-Seedi, H.R., 2021. Marine organisms: Pioneer natural sources of polysaccharides/proteins for green synthesis of nanoparticles and their potential applications. International Journal of Biological Macromolecules, 193, pp.1767-1798.
Taken together:
· The authors would unify the style of the references based on the journal instructions.
· English editing is highly required.
· Please, re-check the punctuation, syntax, and English grammar throughout the manuscript.
Author Response
Dear Reviewer:
Thanks very much for your valuable comments.
Our answers to your comments are as followed:
- We had submitted a graphical abstract with the manuscript submission. And we have revised the graphical abstract and resubmitted it again.
- If the “abstract”means the “graphical abstract”, we have added the most relevant results to the abstract.
- Relevant statement has been added to the section “Introduction”.
- The discussion has been added to the section “2.7. Tx3a.1, Mr3.8 and Vi14b inhibited the proliferation of ID8 cells”.
- This sentence has been revised and is trying to express the urgency for discovery of alternative anti-ovarian cancer drug candidates. Herein, we started this work of finding anti-ovarian cancer conotoxins from Conus venom. So we don’t need to present any examples here.
- The relevant information has been added.
- We found that the figures can be clearly seen in the “Word” format, while the figures in “PDF” format were in low quality. The figures in high quality have been individually submitted.
- Abbreviations has been listed at the end of the article.
- Dr. Ying Fu (corresponding author, associate professor) identified the specimens. This information has been added to section “3.1. Venom Preparation and Peptide Isolation”.
- The error has been revised.
- We have done the statistical analysis (see legend of Figure 9). Statement about statistical analysis has been added to section “3.6. Ovarian Cancer Cell Viability Assay”.
- The conclusion section has been revised.
- Thanks very much for the share of the reference, and we have learned a lot from it.
- The style of the references has been revised based on the journal instructions.
- We have done the English editing (re-check the punctuation, syntax, and grammar) throughout the manuscript.
Thanks again and wish you a good day.
Sincerely,
Dr. Ying Fu
Round 2
Reviewer 3 Report
Comments to authors:
1- The graphical abstract is highly recommended.
2- Please add the most relevant results to the abstract
3- " Conotoxins is a treasure of huge drug resource." Please add examples
4- " This work highlights the role of conotoxin library in subsequent drug discovery for ovarian cancer"; please explain in more details
5- " alternative antitumor drugs are urgently needed for the treatment of ovarian cancer "; could you add some examples
6- " To date, more than 10 contoxins, such as ω-MVIIA, ω-CVID, contulakin-G, conantokin-G, χ-MrIA, α- Vc1.1, κ-PVIIA, μO-MrVIB, μ-SIIIA, CGX-1204, etc., have made it to the clinical research stage "; For which diseases and which phases?
7- Figures are not clear; please improve the quality
8- Please add list of abbreviation
9- Who identifies the specimens?
10- " IC50" should be IC50
11- Could you add statically analysis?
12- The conclusion section should be reduced with focusing on the main outcomes of the study.
13- The authors could benefit from the following reference:
Yosri, N., Khalifa, S.A., Guo, Z., Xu, B., Zou, X. and El-Seedi, H.R., 2021. Marine organisms: Pioneer natural sources of polysaccharides/proteins for green synthesis of nanoparticles and their potential applications. International Journal of Biological Macromolecules, 193, pp.1767-1798.
Taken together:
· The authors would unify the style of the references based on the journal instructions.
· English editing is highly required.
· Please, re-check the punctuations, syntax, and English grammar throughout the manuscript.
Author Response
Dear Reviewer:
Thanks for your valuable comments.
Our answers to your comments are as followed:
- There were several examples in the originally submitted version (30thAug) in Lines 53-59 “To date, more than 7 contoxins, such as ω-MVIIA (chronic pain, marketed), ω-CVID (analgesia, phase IIa), contulakin-G (analgesia, phase I), conantokin-G (analgesia/anti-epileptic, phase Ib), χ-MrIA (analgesia, phase IIa), α-Vc1.1 (analgesia, phase II, terminated), μO-MrVIB (analgesia, phase II), etc., have made it to the clinical research stage [20,21]. Conoxin ω-MVIIA, which targets on N-type human Cav2.2 channel, has been developed to be an FDA-approval analgesic drug Ziconotide (Prialt) for severe chronic pain since its launch in 2004 [22]” for the point “Cone snail venom is a huge drug resources”. Then we had added “Cone snail venom is a huge drug resources, since it is estimated that cone snails can generate up to 1 million different bioactive peptides [23]. However, less than 0.1% of these peptide treasure has been structurally and functionally characterized [15,18,21].” in Lines 59-62 to the revised version (13th Sep). Herein, we had revised this point as required.
- We had discussed this point by adding “In this work, we identified and structurally and biologically characterized the anti-ovarian cancer conotoxins from Conus The result delivered a message that conotoxins, a class of active polypeptides with diverse structure, could be a rich source of anti-ovarian cancer drug candidates. Herein, this study highlights the role of conotoxin library in subsequent drug discovery for ovarian cancer treatment.”in Lines 278-283 to the revised version (13th Sep). Herein, we had revised this point as required.
- We had replied in the revised version (13thSep) that this sentence “Herein, alternative antitumor drugs are urgently needed for the treatment of ovarian cancer” was revised to be “Herein, alternative anti-ovarian cancer drug candidates are urgently needed to be developed for the treatment of ovarian cancer”. We had explained that here we were expressing the the urgency for discovery of alternative anti-ovarian cancer drug candidates. The “alternative candidates” means something that we long for and is the future goal for which we started this work of finding anti-ovarian cancer conotoxins from Conus So we don’t need to present any examples here.
- We had submitted the figures in high quality, and we believe editors will check them.
- The Reference“Yosri, N., Khalifa, S.A., Guo, Z., Xu, B., Zou, X. and El-Seedi, H.R., 2021. Marine organisms: Pioneer natural sources of polysaccharides/proteins for green synthesis of nanoparticles and their potential applications. International Journal of Biological Macromolecules, 193, pp.1767-1798.” is indeed an excellent work from the reviewer. We have learned a lot from this article. However, this review focuses on the green synthesis of metallic, metallic oxides and nonmetallic nanoparticles utilizing extracts/derivatives from marine organisms based on eco-friendly green biogenic procedures and the medicinal and industrial importance of such marine organisms mediated nanoparticles. It clearly showed no relevance to the content of our manuscript. So we had not quoted this reference in the previously revised version (13th Sep) and we will not quote it in this version, as well.
Thanks again and wish you a happy day.
Sincerely,
Ying Fu